



## 1 **On the Generation and Evolution of Internal Solitary Waves in the**
## 2 **Andaman Sea**

Yujun Yu[1], Jinhu Wang[1], Shuya Wang[1], Qun Li[2], Xu Chen[1], Jing Meng[1], Kexiao Lu[1] and Guixia Wang[3]
[1]Key Laboratory of Physical Oceanography, Ocean University of China and Qingdao National Laboratory for Marine Science
and Technology, Qingdao, 266100, China
[2]Polar Research Institute of China (PRIC), Shanghai, 200000, China
[3]College of Mathematical Science, Inner Mongolia Normal University,   Hohhot, 010022,   China
*Correspondence to*: Xu Chen (chenxu001@ouc.edu.cn)
**Abstract.** Internal solitary waves (ISWs) are ubiquitous in the Andaman Sea, as revealed by synthetic aperture radar (SAR)
images, but their generation mechanisms and corresponding influencing factors remain unknown. Based on a nonhydrostatic
two-dimensional model, the generation of ISW packets along a transect of a channel lying between Batti Malv Island and Car
Nicobar Island is investigated. Additionally, the influences of topographic characteristics, seasonal stratification variables and
tidal forcings are analysed through a series of sensitivity runs. The simulated results indicate that bidirectional rank-ordered
ISW packets are generated by the nonlinear steepening of internal tides. An east-west ISW asymmetry is observed, which is
attributed to distinct topographic characteristics. The surrounding sills are also capable of generating internal wave beams,
which modulate the intensity of ISWs. However, the topographic structure of the west flank of the ridge mainly contributes to
the suppression of westward ISWs, which decrease the modulating effect of internal wave beams. During spring tide, the
generation of ISWs is enhanced. Under neap tide, ISWs are weak, and the east-west ISW asymmetry is less obvious. Moreover,
seasonally varied stratification only has a minor effect on the generation and evolution of ISWs.
**1. Introduction**
Internal solitary waves (ISWs) are a ubiquitous phenomenon in marginal seas (Jackson, 2007). Accompanied by strong
horizontal and vertical currents, large-amplitude ISWs can propagate a long distance from their generation sites, while keeping
their waveform nearly invariant (e.g., Alford et al., 2010; Huang et al., 2014; 2016; Lien et al., 2012; 2014). As a result, they
carry the potential to damage offshore engineering structures (Xu et al., 2012) and considerably impact nutrient transport
systems (Dong et al., 2015). When ISWs shoal onto slops or shelf topographies, they breakdown and therefore induce enhanced
turbulent mixing (Vlasenko et al., 2002; Vlasenko and Stashchuk, 2007; Moum et al., 2007; Sutherland et al., 2013; Lamb,
2014; Jones et al., 2020).
Overall, the generation of ISWs is closely related to tide-topography interactions. There are several mechanisms for the
generation of ISWs. One is the nonlinear steepening of internal tides (Lee and Beardsley, 1974). For moderate tidal currents,
linear internal waves are first generated over topography. As they radiate away from such landforms, they gradually evolve





into ISWs (Farmer et al., 2009; Li and Farmer, 2011; Li, 2014; Buijsman et al 2010; Alford et al., 2015). The second mechanism
is the formation of Lee waves (Maxworthy, 1979). In the presence of strong tidal currents, depression waves are formed on
the leeward side of topography by supercritical tidal flow. As the tidal flow weakens, the depression wave starts to propagate
upstream and then develops into ISWs near topography. The third mechanism is the "local" formation of ISWs induced by the
interaction of internal wave beams with the thermocline (Gerkema, 2001). In addition to the above mechanisms, ISWs can be
generated by other mechanisms, e.g., local collapse events (Maxworthy 1979), internal hydraulic jumps (Cummins et al., 2006)
and upstream influences (Raju et al., 2021).
The Andaman Sea (AS) is located in the northeast Indian Ocean. Due to the presence of shallow ridges, strong stratification
tendencies and tidal currents, the AS is regarded as a hotspot of ISW generation. Since the oceanography survey of Perry and
Schimke (1965), numerous studies of ISWs through in situ measurement, remote sensing and numerical modelling have been
carried out (e.g., Osborne and Burch, 1980; Alpers et al., 1997; Jackson et al., 2012; Jensen et al., 2020; Magalhaes et al., 2020;
Raju et al., 2021). There are several types of ISWs active in the AS, including rank-ordered mode-1 wave packets and high
mode waves (da Silva and Magalhaes, 2016; Magalhaes and da Silva, 2018; Raju et al., 2019). ISWs in the AS are mainly
generated in the western island chain, which includes the Nicobar archipelago and the Andaman Islands (Raju et al., 2019;
2021; Jensen et al., 2020; Magalhaes et al., 2020). With a steep topography and a high degree of barotropic to baroclinic energy
conversion (Mohanty et al., 2018), multiple channels in the Nicobar archipelago are potential generation points of ISWs (Raju
et al., 2019; 2021; Jensen et al., 2020). In Fig. 1, bidirectional ISWs propagating from submarine ridge R0 located at 92.83 °E,
8.94 °N are visible in the channel between Car Nicobar Island and Batti Malv Island. Using SAR imagery from TerraSAR-X
in the Andaman Sea between 8°N~10°N, Magalhaes and da Silva (2018) showed the pattern of ISWs in that region and noted
that the generation of ISWs is attributed to beam-pycnocline interactions. A study by Magalhaes et al. (2020) revealed that the
topographic characteristics of the 10°N channel play an important role in the generation of secondary ISWs. Through 3-D
nonhydrostatic modelling, Raju et al. (2021) described the generation mechanisms of ISWs in the channel between Car Nicobar
Island and Batti Malv Island and the channel between Batti Malv Island and Chowra Island (see details in Raju et al., 2021).
As mentioned above, the mechanism of ISW generation in the channel in Fig. 1 is still controversial, and factors that affect
ISW generation therein have not been adequately explored, which provides motivation for this work.
In this work, to explore the generation and evolution of ISWs from a ridge located at 92.83 °E, 8.94 °N, numerical simulations
are performed. In addition, several sensitivity experiments are carried out to examine the corresponding influencing factors.
The remainder of this paper is organised as follows. Section 2 presents the modal setup and the nondimensional parameters
considered in ISW dynamics. The result of the standard run is presented in Section 3. The impacts of topography, tidal forcing
and seasonal stratification on the generation and evolution of ISWs are discussed in Section 4. Finally, the paper is summarised
in Section 5, along with some further discussions.

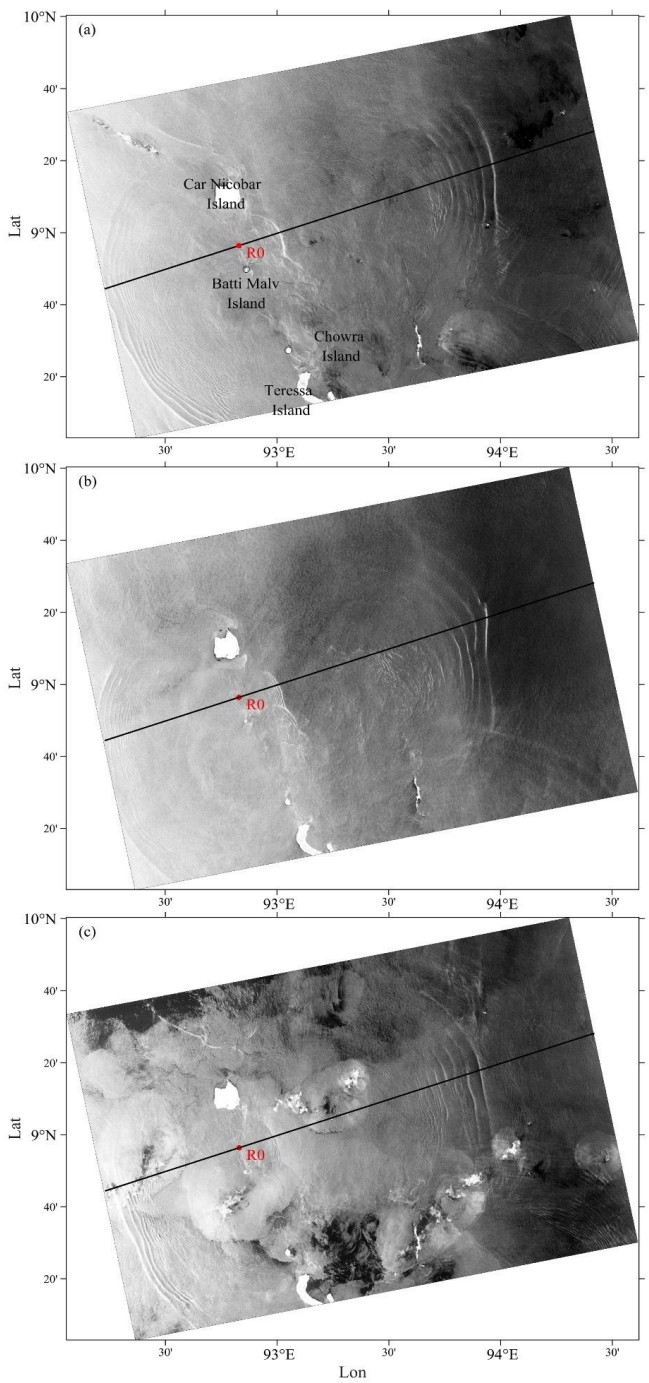


**Fig. 1 Sentinel-l ASAR image of the Bay of Bengal acquired at (a) 12:00 UTC on 12 Apr 2017, (b) 12:00 UTC on 2 Mar 2018 and (c)**

**12:00 UTC on 1 May 2018. The topographic section in the numerical simulation is marked with black lines.**



**2. Methodology**
**2.1 Model setup**
In this work, a fully nonlinear nonhydrostatic model, namely, the Massachusetts Institute of Technology General Circulation
Model (MITgcm) (Marshall et al., 1997), is employed. For simplification, a 2-D ($x$-$z$ plane) configuration is considered. The
horizontal direction resolution is 500 m in the central region, which is sufficient to scrutinize detailed wave structures and
comparable to that used in previous studies (Buijsman et al, 2010; Li, 2014; Vlasenko et al., 2018); it is gradually stretched to
1 km towards the open boundaries. There are 140 uneven layers in the vertical direction, with thicknesses increasing from 10
m near the surface to 50 m near the bottom. The time step is set to 15 s, which satisfies the Courant-Friedrichs-Lewy (CFL)
condition, and results are output every 10 minutes. The Coriolis frequency is $2.28 \times 10^{-5}$ s$^{-1}$, corresponding to a latitude of 9°N.
The horizontal viscosity is set to $v_h$=25 m$^2$/s to suppress grid-scale noise (Legg and Huijits, 2006), and the horizontal diffusivity
is set to $\kappa_h$=10$^{-3}$ m$^2$/s. The PP81 scheme (Pacanowski and Philander, 1981) is applied to calculate vertical viscosity and
diffusivity.
$$v = \frac{v_0}{(1+\mu Ri)^n} + v_b \qquad (1)$$
$$\kappa = \frac{v}{(1+\mu Ri)} + \kappa_b \qquad (2)$$
where $Ri$ is the Richardson number, $N$ is the buoyancy frequency, and $v_b$=10$^{-5}$ m$^2$/s and $\kappa_b$=10$^{-5}$ m$^2$/s are the background
viscosity and diffusivity values, respectively. Following previous studies, we set $v_0$=1.5×10$^{-2}$ m$^2$/s, $\alpha$=1 and $n$=1 (e.g., Vlasenko
et al., 2010; 2012; Min et al., 2019, Wang et al., 2020). Sponge layers are added to the east and west boundaries to avoid the
reflection of baroclinic waves. In addition, the no-slip condition represents the bottom boundary. The model is forced by adding
a force to the right-hand side of the momentum equations (Vlasenko et al. 2010; 2012; Guo et al., 2011). All the simulations
are operated for 7 days.

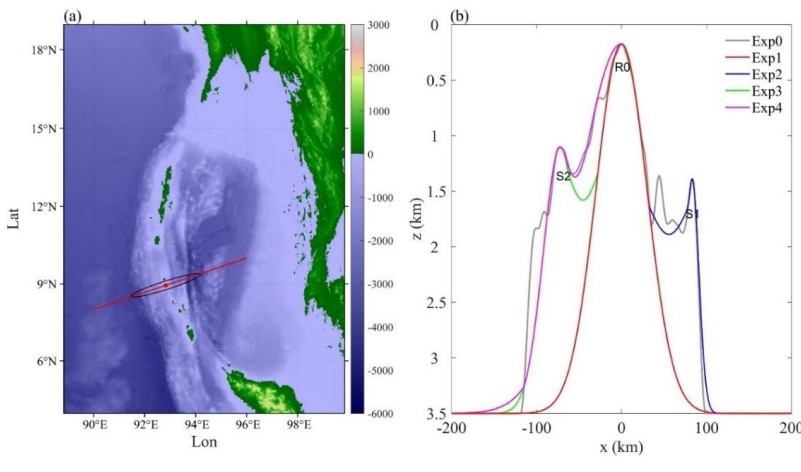




**Fig. 2 (a) The bathymetry of the AS. The black line is the tidal ellipse of M2 at R0, and the red line is the transect of the 2-D domain**
**of the simulation corresponding to (b). Sills S1 and S2 surround ridge R0, which may have an effect on the generation of ISWs from**
**R0. It should be noted that the west flank of R0 has distinct characteristics, with a smoother slope and shallower depth than the east**
**flank.**
Bathymetry data are extracted from the ETOPO1 dataset along the transect shown in Fig. 2, which has a direction that is
generally consistent with the propagating direction of ISWs (Fig. 1). The maximal depth is set to 3500 m. In addition to crest
R0, the realistic topography has several sills (Fig. 2b). To explore the impact of topography on ISW generation and evolution,
several sensitivity experiments are carried out (Exps1-4). In Exp1, a fitted Gaussian ridge is used. In Exp2 and Exp3, the fitted
Gaussian ridges of R0+S1 and R0+S2 are considered to explore the role of small sills. To further discuss the effect of distinct
topographic structures, the modified topography R0+S2 is used in Exp 4.

**Table 1 Data on the major axis of the tidal ellipse of eight principal tidal constituents**

|  | $M_2$ | $K_2$ | $N_2$ | $S_2$ | $K_1$ | $O_1$ | $P_1$ | $Q_1$ |
|---|---|---|---|---|---|---|---|---|
| **Amplitude (cm/s)** | 42.7 | 6.5 | 8.4 | 22.1 | 5.6 | 2.2 | 1.7 | 0.3 |
| **Phase (°)** | 17.8 | 19.1 | 19.6 | 17.6 | 16.3 | 15.1 | 16.1 | 12.2 |


The barotropic tidal currents are extracted by the Oregon State University Tidal Inversion Software (OTIS, Egbert and
Erofeeva, 2002) global atlas (TPXO7.2) at 92.83 °E, 8.94 °N. As shown in Table 1, semidiurnal tides are predominant in the
Andaman Sea, which has also been noted by previous studies (e.g., da Silva and Magalhaes, 2016; Raju et al., 2019). In contrast,
the contribution of diurnal tides can be negligible (Table 1). In a standard run, the $M_2$ tidal force is imposed. To explore the
variation in ISW generation during spring and neap tides, the $M_2$ and $S_2$ tides are taken into consideration in sensitivity runs
(Fig. 3).

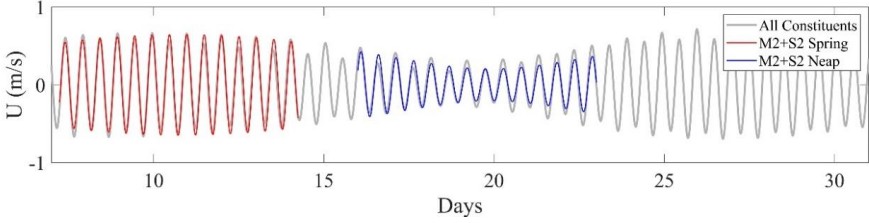


**Fig. 3 Tidal currents at R0 (92.83 °E, 8.94 °N) derived by OTIS.**
Horizontally uniform stratification is employed in this work and extracted from the World Ocean Atlas 2018 (WOA18) dataset.
The maximum buoyancy frequency is located at 90 m and has a value of 0.021 $s^{-1}$ (Fig. 4). Considering that stratification could
affect the generation of ISWs, summer and winter stratifications are used in sensitivity runs (Exp 7-8) for corresponding
exploration.




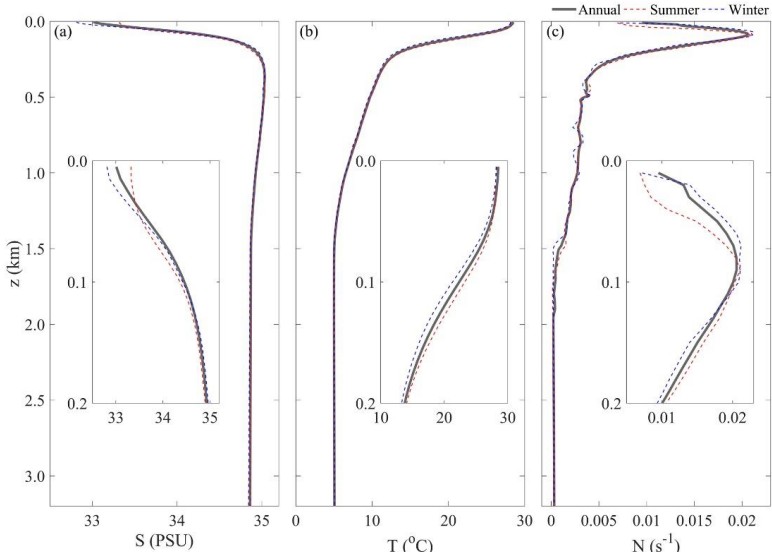


**Fig. 4 Initial (a) salinity, (b) temperature and (c) buoyancy frequency profiles.**

**Table 2 Configurations of sensitivity experiments**

| Name | Topography | Tidal Force | Stratification |
|------|-----------|-------------|----------------|
| Exp0 | R0+S1+S2 | $M_2$ | Annual |
| Exp1 | R0 | $M_2$ | Annual |
| Exp2 | R0+S1 | $M_2$ | Annual |
| Exp3 | R0+S2 | $M_2$ | Annual |
| Exp4 | Modified R0+S2 | $M_2$ | Annual |
| Exp4 | R0+S1+S2 | $M_2+S_2$ spring tide | Annual |
| Exp5 | R0+S1+S2 | $M_2+S_2$ neap tide | Annual |
| Exp6 | R0+S1+S2 | $M_2$ | Summer |
| Exp7 | R0+S1+S2 | $M_2$ | Winter |

**2.2 Non-dimensional parameters**
As suggested by previous studies (Buijsman et al., 2010; Guo et al., 2011; Vlasenko et al., 2012), several nondimensionless
parameters control the generation of ISWs:
(1) The tidal excursion length is represented by $\sigma = U/(L\omega)$, where L is a horizontal topographic length scale, and $U$ is the
amplitude of the tide current. When $\sigma \ll 1$, only internal waves with a given tidal frequency are generated. However, when
$\sigma \gg 1$, internal Lee waves along with higher harmonics are formed (Bell, 1975). When $\sigma \sim 1$, a "mixed tidal lee wave" regime
appears since the time derivative and advection provide comparable contributions (Nakamura et al., 2000; Vlasenko et al.,

121    2005).

(2) The slope criticality parameter is represented by $\gamma = (dh/dx)/\alpha$, where $dh/dx$ is the topographic slope and $\alpha =$
$\sqrt{(\omega^2 - f^2)/(N^2 - \omega^2)}$ is the ray slope of the internal waves. Topographies where $\gamma < 1$, $\gamma = 1$ and $\gamma > 1$ are defined
as "subcritical", "critical" and "supercritical", respectively.
(3) The topographic Froude number is represented by $Fr_t = U/(N_{max}H)$, where $N_{max}$ is the maximum value of $N(z)$ and $H$ is
the topographical height (Legg and Huijts, 2006; Legg and Klymak, 2008). This parameter describes the blocking and
nonlinear hydraulic effect caused by topography. When $Fr_t < 1$, the topography affects the flow, resulting in the occurrence
of blocks. In contrast, when $Fr_t > 1$, the flow is not affected by the topography. When $Fr_t \sim 1$, tidal flows over a given
topographical setting share common properties in terms of their harmonic oscillation and the presence unsteady Lee waves,
which indicate mixed-lee waves (Nakamura et al. 2000).
(4) The internal Froude number is defined as the ratio of barotropic velocity $U$ to the internal wave speed $c$, i.e., $Fr_w = U/c_i$.
The linear internal wave speed $c$ is calculated by solving the boundary value problem (BVP) (Gill, 1982), along with the
prescribed boundary condition $\phi(0) = \phi(-H) = 0$.
$$\frac{d^2\phi(z)}{dz} + \frac{N^2(z)}{c^2}\phi(z) = 0,$$    (3)
Waves behave linearly for $Fr_w \ll 1$ but become nonlinear when $Fr_w \sim 1$.

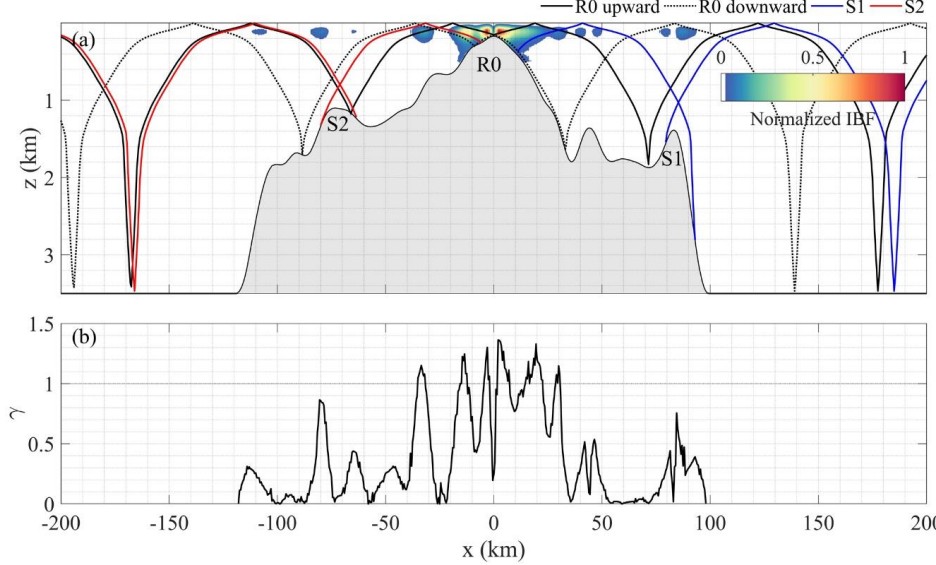

**Fig. 5 (a) Normalised IBF for realistic bathymetry. Red, blue and green lines mark the ray paths from R0, S1 and S2, respectively,**
**which are calculated by the linear dispersion relation $dz/dx = \sqrt{(\omega^2 - f^2)/(N^2 - \omega^2)}$. (b) Criticality parameter for the realistic**
**topography.**
Potential generation points of internal waves can be predicted by calculating the internal body force (IBF) as follows:





$$IBF = \rho_0 U h(x) z \frac{N_z^2}{\omega}\left[\frac{1}{h(x)}\right]_x.$$ (4)
where $U$ and $\omega$ are the amplitude and frequency of barotropic tides, respectively (Baines, 1973; Li, 2014; Vlasenko et al.,

143 2018).

The dominant generation point of internal waves is R0, as shown in Fig. 5(a). The amplitude of the barotropic current on R0
is approximately 0.4 m/s. According to Table 3, the topographic Froude value is 0.286 for the realistic topography, indicating
the occurrence of blocking. When $\delta \ll 1$ and $\gamma > 1$, ridge R0 falls within regime 5 according to Garrett and Kunze (2007),
featuring internal waves generated at higher harmonic tidal frequencies.

**Table 3 Physical parameters for ISWs generated in the Andaman Sea.**

| $U_{max}$(m/s) | $c_1$(m/s) | $H_{R0}$(m) | L(km) | $\omega$(rad/s) | $N_{max}$(rad/s) | $\delta$ | $Fr_t$ | $Fr_w$ | $\gamma$ |
|---|---|---|---|---|---|---|---|---|---|
| 0.43 | 1.05 | 3326 | 30 | 1.41×10-4 | 2.6×10-2 | 0.095 | 0.286 | 0.41 | 1.36 |


**3. Standard run**

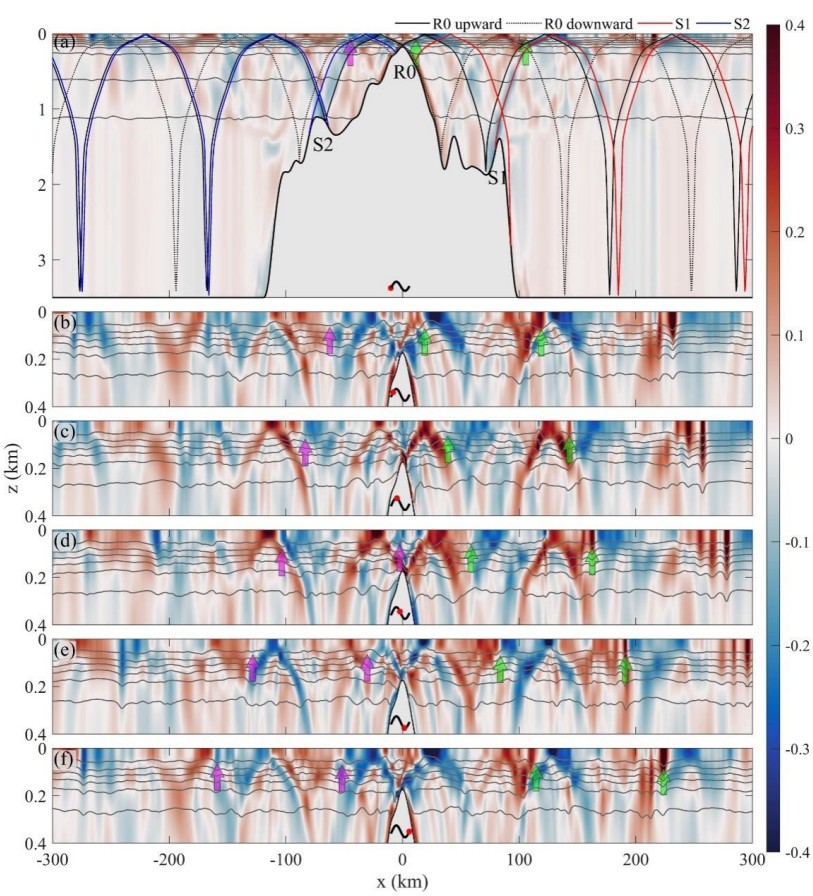

**Fig. 6 Snapshots of horizontal baroclinic velocities (shadings, unit: m/s) and isotherms (grey contours) during a tidal cycle. Phase of each snapshot is denoted by a red dot. Characteristic rays of internal waves are marked by coloured lines. Green and violet arrows denote eastward- and westward-propagating ISWs, respectively.**

Fig. 6 displays snapshots of the baroclinic velocity along with isotherms within one tidal cycle. Overall, the generation points
of the internal wave beams are consistent with their predicted IBFs in Fig. 5(a). The internal wave beams (black lines in Fig.
6a) are supercritically generated from R0, and they propagate obliquely according to the dispersion relation in accordance with
regime 5 proposed by Garrett and Kunze (2007). Meanwhile, internal wave beams radiating from S1 and S2 are also visible
(blue and red lines in Fig. 6a). Far away from the topographic landforms, beam-like structures gradually become invisible,
whereas low modes are dominant because high-mode waves are always dissipated locally due to a strong vertical shear force
(Pickering and Alford, 2012). In addition, at the start of flood or ebb tides, depression waves on the leeward side of the
topographic feature are visible. As they propagate downstream, the depression waves evolve into rank-ordered ISW packets
(coloured arrows in Fig. 6). Fig. 6 indicates that the eastward ISWs are more energetic than the westward ISWs. The Hovmuller
diagram of $T(x, t)$ at $z$=100 m is shown in Fig. 7.  ISW packets are indicated by diagonal fingers, with a speed of $c_1$=2.51 m/s,



which agrees with that resulting from Equation (3). In addition, internal mode-2 and mode-3 waves are also detected in Fig. 7,
with speeds of $c_2$=1.49 m/s and $c_3$=0.83 m/s, respectively, which are consistent with theoretical values. However, these high-
mode internal waves do not evolve into ISWs.

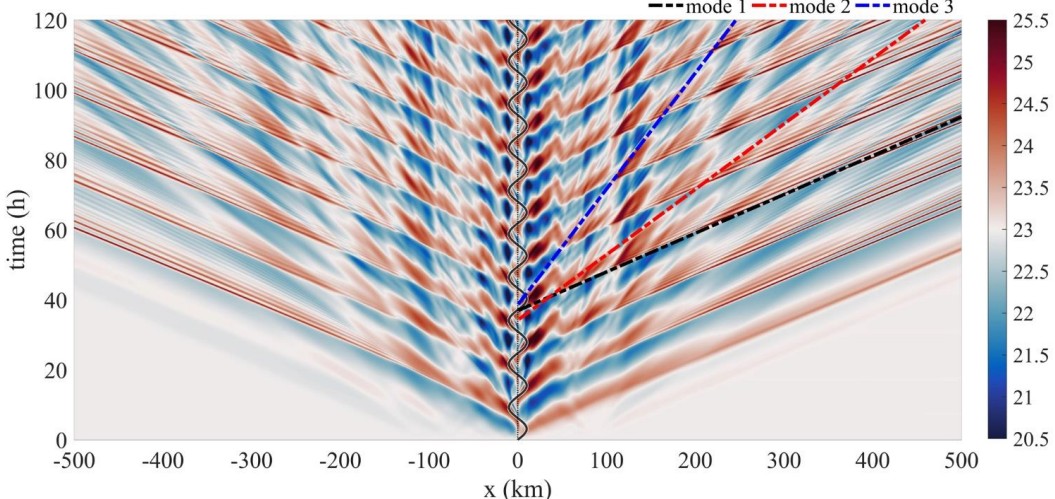


**Fig. 7 Hovmuller diagram of $T$ (shading, unit:°C) at $z$=100 m for Exp1. The black line is the barotropic tide current at R0, and the**
**black, red, blue dotted-dashed lines are the first-, second- and third-mode internal waves, respectively.**
By tracing the waves back to the generation sites, it can be found that the generation of ISWs is related to the time at which
the tidal flow pattern changes its direction. Fig. 6 and Fig. 7 reveal that ISWs trace back to depression waves (marked by
coloured arrows in Fig. 6), which originate from downstream tidal flow. The depression waves downstream propagate
approximately 100 km before they evolve into ISWs. Therefore, the generation mechanism of bidirectional ISWs is the
nonlinear steepening of internal tide mechanisms (Lee and Beardsley, 1974; Bujisman et al. 2010). The flood tide moves from
west to the east of R0; a depression wave propagates away from topography, and it evolves into a rank-ordered ISW packet
due to its nonlinearity. Moreover, no ISWs evolved from elevation waves from R0.





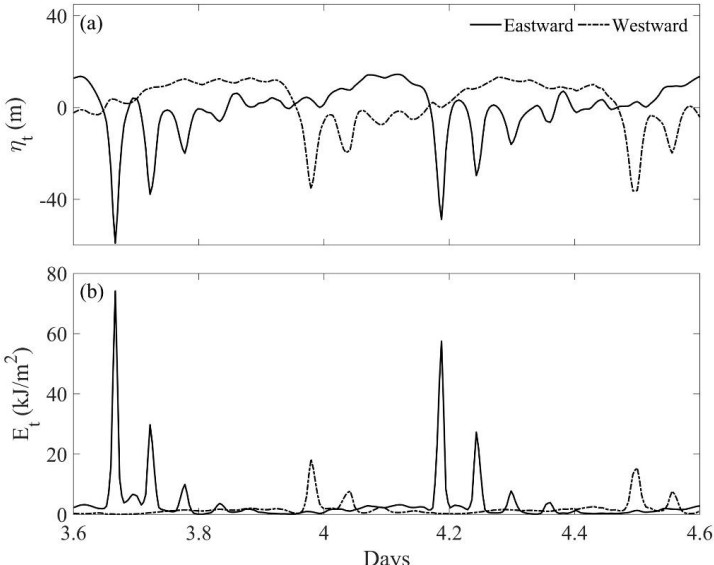


**Fig. 8 (a) Vertical displacement of the 9.8°C isotherms and (b) depth integrated energy for eastward and westward ISWs at *x*=±300**

**km.**
The characteristics of ISWs can be examined through the times series of vertical displacement and energy. As shown in Fig.
8a, overall, both eastward- and westward-moving ISWs behave as rand-ordered wave packets, agreeing with the results derived
from SAR images (Fig. 1). However, the eastward waves have a larger amplitude (59.1 m) than the westward waves (40.3 m)
and involve more secondary waves in the packets. This result indicates asymmetric ISW generation. Moreover, the energy of
ISWs is calculated as the sum of baroclinic kinetic energy and available potential energy (Buijsman et al., 2010) as follows:
$E_k = \frac{1}{2}\rho_0 \int_{-H}^{0}(u^2 + w^2)dz$ (5)
$E_p = \frac{1}{2}\rho_0 \int_{-H}^{0} \frac{b^2}{N^2} dz$ (6)
where $u$ is the baroclinic velocity, $w$ is the vertical velocity, buoyancy $b = -g\rho'/\rho_0$, and $\rho_0$ and $\rho'$ are the background and
perturbation density, respectively. The energy of the eastward ISWs reaches 74.2 kJ/m$^2$, which is basically four times larger
than that of the westward ISWs. These results indicate that although ISWs are observed in both the Bay of Bangel (BoB) and
the AS, ISWs in the two regions actually have different intensities. ISWs in the AS are more energetic than those in the BoB.





## 4. Sensitivity experiments

### 4.1 Topographic features

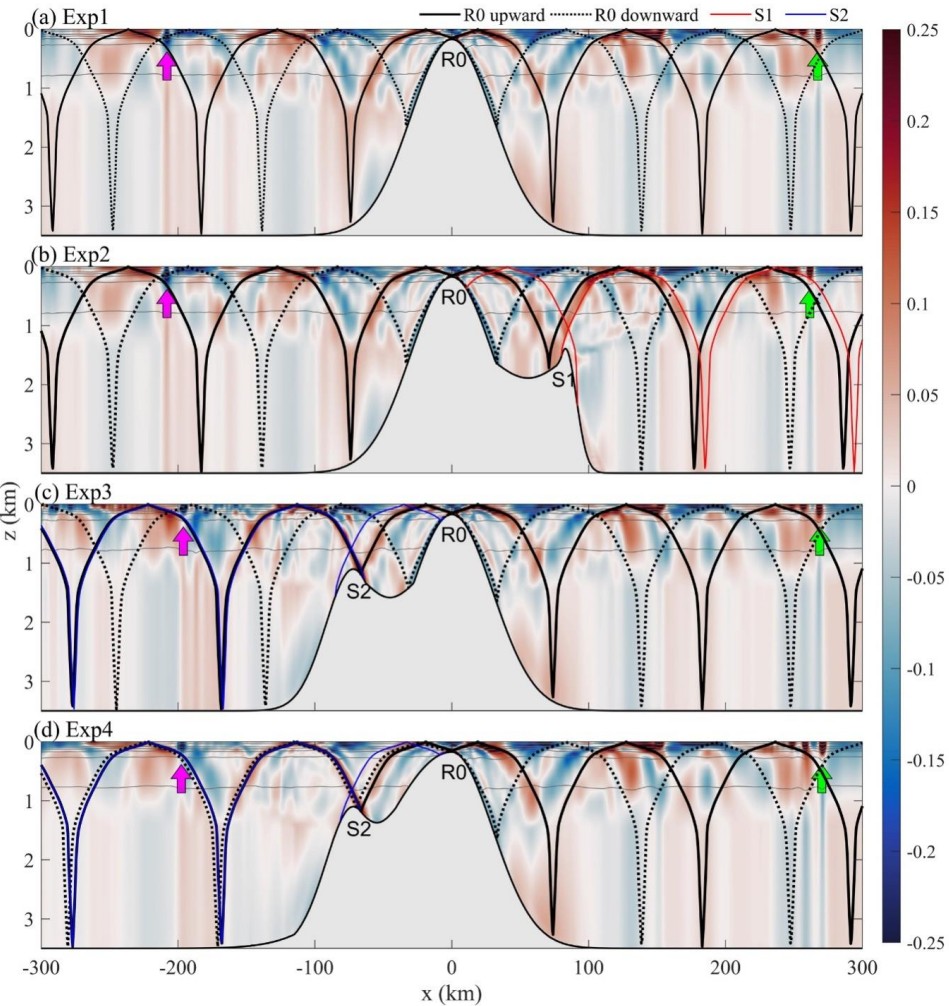

**Fig. 9 Snapshots of horizontal baroclinic velocities (shadings, unit: m/s) and the associated isotherms (grey contours) of Exp1~4 at 91 h. Characteristic rays of internal waves are denoted by coloured lines. Green and violet arrows denote eastward- and westward-propagating ISWs, respectively.**

The asymmetry of ISWs on the west and east sides of the landform is illustrated in Section 3; this asymmetry may be related to the asymmetric features of the topography. Therefore, sensitivity runs Exp1-4 are carried out to explore the impact of topography on ISW generation. The topography utilized in Exp1 only features the Gaussian fitting of R0 in Fig. 9b, while the topographies employed in Exp2 and Exp3 fit R0+S1 (Fig. 9c) and R0+S2 (Fig. 9d), respectively. In Exp1 (Fig. 9a), because the ridge is supercritical, both upward and downward internal tidal beams are observed. Consistent with Exp0, no ISWs were generated near the landform. Rank-ordered ISW packets emerge approximately 100 km away from ridge crest R0. The





generation mechanism of bidirectional ISWs is also the nonlinear steepening of internal tides. In Exp2 and Exp3 (Fig. 9b-c),
in addition to the beams generated by R0, beams radiating from S1 and S2 are also detected, resulting in a wave field exhibiting
asymmetric features.

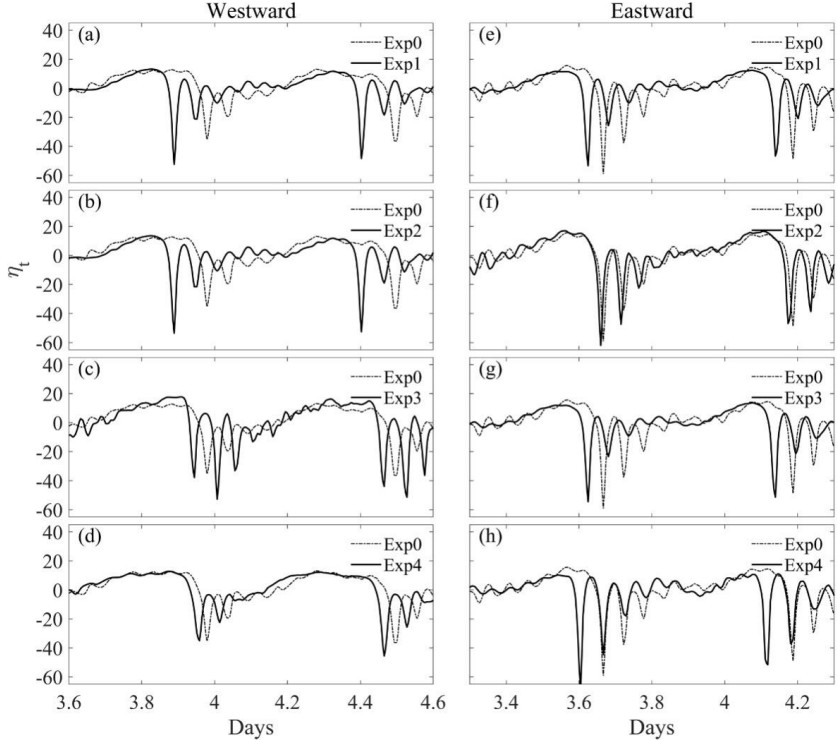


**Fig. 10 Vertical displacement of the 9.8°C isotherms for the (a)-(d) westward and (e)-(h) eastward ISWs at $x=\pm300$ km.**

A quantitative comparison of the bidirectional ISWs produced in these sensitivity runs is shown in Fig. 10. First, the
bidirectional ISWs in Exp1 are symmetrical (Fig. 10a and 10e), indicating that the asymmetry of ISWs could be attributed to
S1 and S2. The west flank of R0 in Exp2 is similar to that in Exp1, which results in the same pattern and different patterns of
internal waves in the west and east regions of R0, respectively (Fig. 9a and b). Therefore, similar east flanking waves relative
to R0 were observed in Exp1 and Exp3 (Fig. 9a and c). As the internal wave beams from S1 and S2 reflect off the supercritical
slope of R0, they were blocked by ridge R0. Moreover, the westward (eastward) ISWs of Exp2 (Exp3) in Fig. 10b (Fig. 10c)
are similar to those of Exp1. Above all, when ebb (flood) tidal currents flowed over R0 in Exp2 (Exp3), the upstream dynamic
process had a minor effect on the depression waves generated from R0. Hence, it provides further evidence that ISWs are
generated due to the nonlinear steepening of internal tides.
In Fig. 10f, the eastward ISWs of Exp2 are similar to the standard run Exp0, which indicates that sill S1 has a major effect on
the evolution of eastward ISWs. In Fig. 9(b), the eastward upward internal wave beams from R0 (shown by black solid lines),

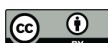



which are reflected twice, are nearly in-phase with the internal wave beams from the west flank of S1 (red lines). In that case,

the amplitude of the internal wave is enhanced by the modulating effect of those internal waves, resulting in the strengthening

of the nonlinear evolution of eastward ISWs.

In Fig. 9c, the modulating effect of internal waves from upward wave beams from R0 and S2 also contributes to the

enhancement of westward ISWs (Fig. 10c). However, interestingly, considering the above effect, the westward rank-ordered

ISW packets of the standard run (dotted-dashed line in Fig. 10c) are instead weakened. Hence, another sensitivity experiment

Exp4 is carried out, and the topography of the landform has a modified west flank of ridge R0 in contrast to that of Exp3,

which is similar to its realistic topography. In Fig. 10d, the suppression of the evolution of ISWs indicates that the west flank

of ridge R0 has a major suppressive effect on ISW generation. Then, the suppressive effect of the west flank of R0 decreases

the modulating effect of westward internal waves from R0 and S2. It should be noted that the slight difference in slope on the

west slope between Exp1 and Exp4 results in the difference between eastward internal waves shown in Fig. 10(h), which has

not been considered in a previous study.

## 4.2 Tidal forcing

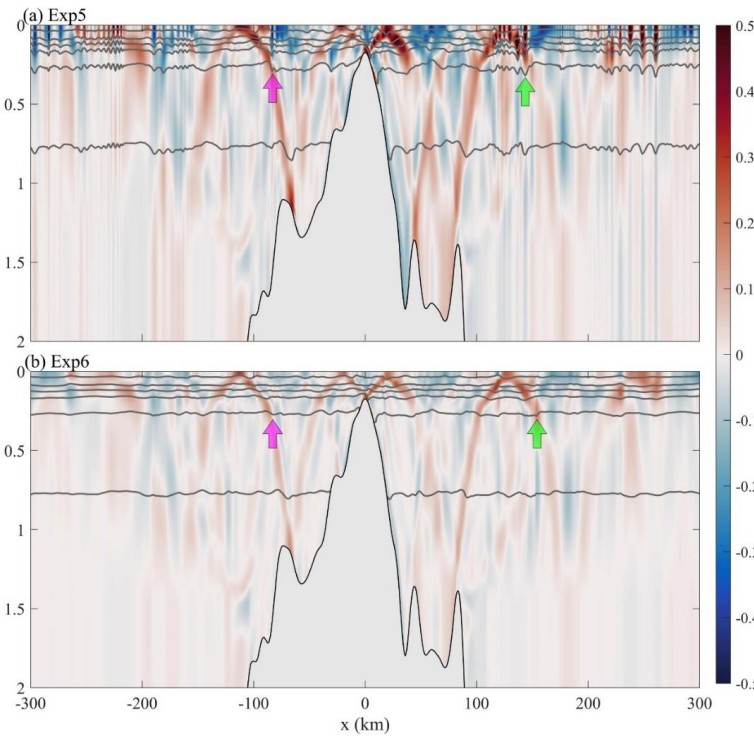

**Fig. 11 Snapshots of horizontal baroclinic velocities (shadings, unit: m/s) and isotherms (grey contours) of Exp 4~5 at 91 h.**



Tidal currents vary during spring and neap tides in the AS, which could affect ISW generation. Therefore, Exp5-6 are carried
out. During spring tide, internal tidal beams are obviously enhanced (Fig. 11a). However, during neap tide, no pronounced
ISWs are identified. The topographic Froude numbers for the simulated spring and neap tides are 0.428 and 0.155, respectively,
which fall within the same regime as Exp0 according Garrett and Kunze's (2007) regime. Snapshots of the wave field
generating during a tidal cycle for Exp5 are shown in Fig. 12. Depression waves evolve into rank-ordered ISW packets, which
are also generated from the start of flood or ebb tides. The generation of bidirectional ISWs is also attributed to the nonlinear
steepening of internal tides. Meanwhile, the modulating effect of internal waves mentioned in Section 4.1 is enhanced
(coloured box in Fig. 12). Moreover, more waves are included in the wave packets during spring tide.

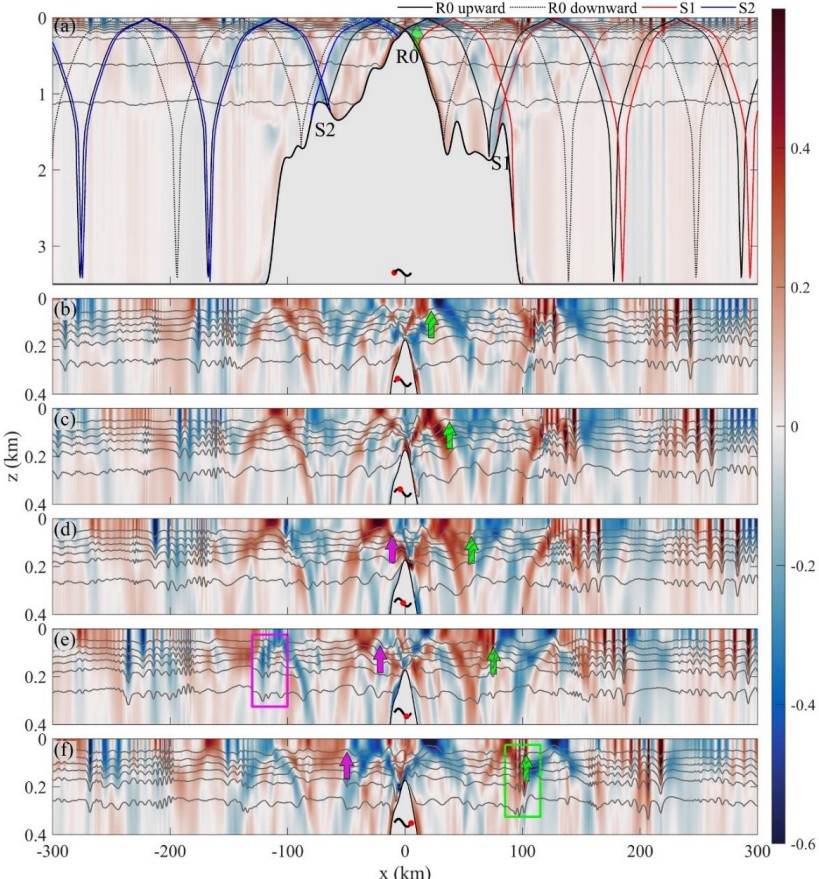

**Fig. 12 Snapshots of the spring tidal force in Exp5, which is similar to Fig. 6.**
The vertical displacement at $z$=500 m and the depth-integrated baroclinic energy gradient are shown in Fig. 13. During spring
tide, the rank-ordered ISW packets are enhanced. The westward leading ISW is strengthened with an amplitude of 54 m, which
is 14 m larger than that induced during the standard run. The amplitude of the eastward leading ISW (76 m) is slightly larger
than that of the standard running Exp0, whereas the secondary ISWs of the wave packets are much stronger than those in Exp0





(Fig. 13a and b). Overall, the baroclinic energy of the leading wave is maximized in rank-ordered ISWs, which reaches 215.9
kJ/m$^2$. However, during neap tide, the amplitudes of both eastward and westward ISWs decrease to 20 m, and the baroclinic
energies are 7.2 kJ/m$^2$ and 4.2 kJ/m$^2$, respectively. Moreover, during neap tide, fewer waves are included in the rank-ordered
wave packets.

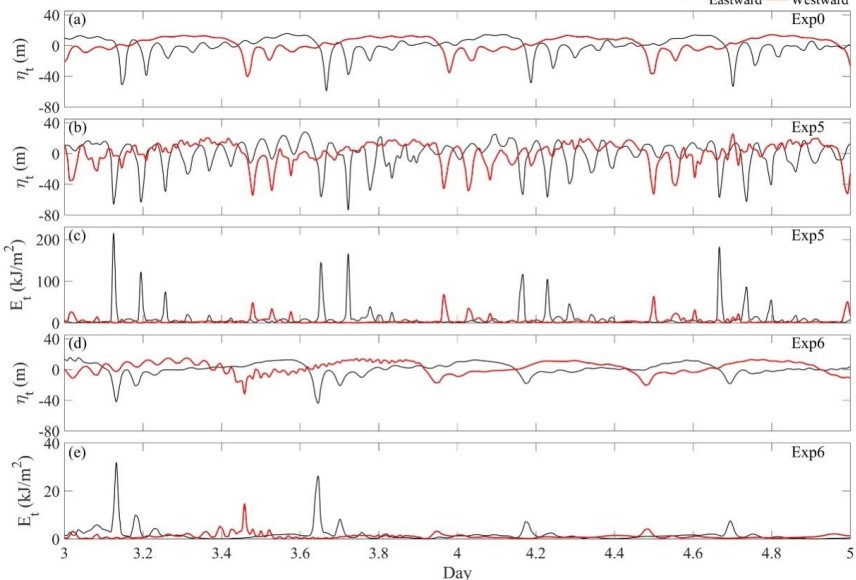


**Fig. 13 Vertical displacement of the 9.8°C isotherms in (a) Exp0, (b) Exp5 and (e) Exp6 and depth integrated energy in (c) Exp5 and**
**(e) Exp6 for eastward (black lines) and westward (red lines) ISWs at $x=\pm300$ km.**
**4.3 Seasonal Stratification**

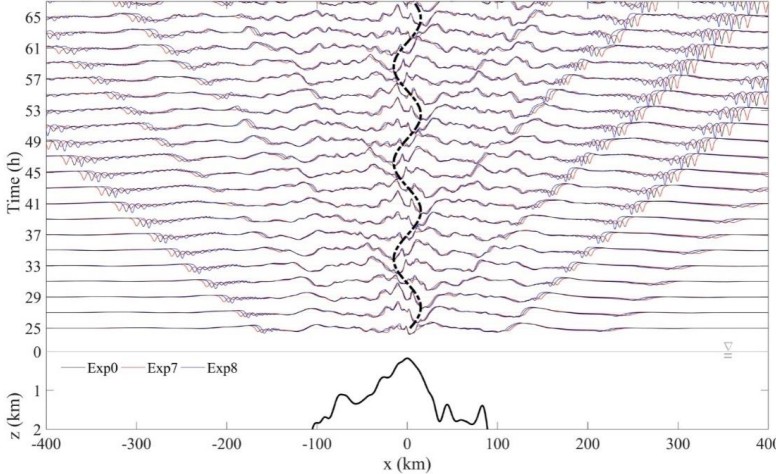




**Fig. 14 Comparison of isotherms at a vertical level of 10 (z=95 m) between Exp0, Exp7 and Exp8 in the upper frame, where black,**
**red and blue lines are Exp0, Exp7 and Exp8, respectively. The lower subplot is bathymetry at z=0−2 km.**
To investigate the influence of seasonal stratification on the generation and evolution of ISWs, Exp7 and Exp8 are carried out,
and comparisons of the results are shown in Fig. 14. Near the ridge crest, the isothermal displacements are almost the same for
Exp0 and Exp7-8. However, as depression waves evolve, ISWs in Exp7 are slightly faster than those in Exp0 and Exp8 shown
in Fig. 14. The phase speeds of the mode-1 internal wave are $c_1$=2.57 m/s and 2.43 m/s in summer and winter, respectively. In
Fig. 15, the slightly strong pycnocline in summer results in a larger amplitude (61 m) and a greater amount of baroclinic energy
(104 kJ/m$^2$) than that in winter. However, there are 3~4 ISWs in eastward rank-ordered packets in each simulation. On the
other hand, seasonal stratifications have no effect on the evolution of westward ISWs, which might be due to the suppressive
effect of ISWs by the topographic structure. Generally, due to slight differences in the buoyancy frequency, seasonal
stratification only has a minor effect on the generation and evolution of ISWs.

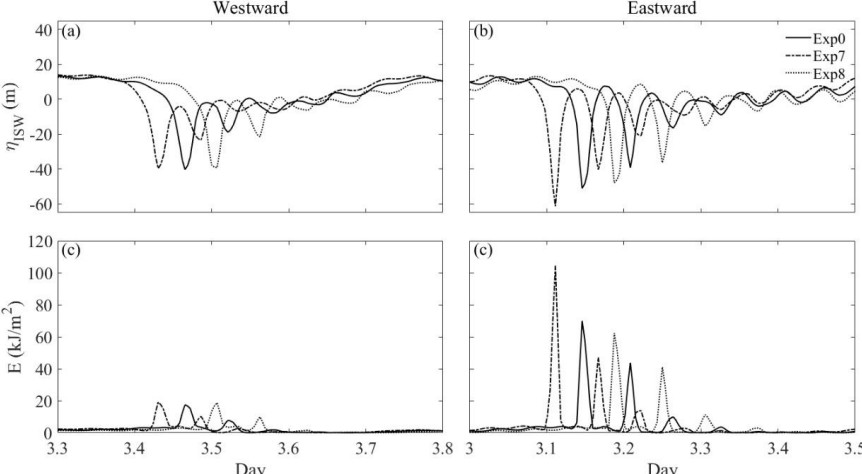


**Fig. 15 (a)-(b) Vertical displacement of the 9.8°C isotherms and (c)-(d) depth integrated energy for eastward and westward ISWs of**
**Exp0, Exp7 and Exp8 at $x$=±300 km, respectively.**
**5. Summary and discussion**
To investigate the asymmetry of bidirectional ISWs in the Andaman Sea, the fully nonlinear nonhydrostatic 2-D MITgcm was
applied. In the standard run, the M$_2$ tidal constituent forcing and realistic topography of the 8.94°N transect at the Nicobar
archipelago in the Andaman Sea were configured. A series of sensitivity experiments were carried out to explore the influences
of topography, tidal flow and seasonal stratification on the generation and evolution of ISWs, and the main conclusions of this
study are listed as follows.



(1) Mode-1 rank-ordered ISW packets in this region are mainly generated by semidiurnal barotropic tides. With the subcritical
tidal flow extracted from the OTIS, the generation mechanism of bidirectional ISWs is the nonlinear steepening of internal
tides. Moreover, bidirectional ISWs exhibit an asymmetrical feature.
(2) Distinct topographic characteristics play an important role in the asymmetry of bidirectional ISWs. Enhanced amplitude
by internal wave beams from R0+S1 and R0+S2 reinforces the evolution of bidirectional ISWs. However, the topographic
features of the west flank of R0 decrease the energy of internal waves, resulting in the suppression of the evolution of westward
ISWs. This indicates the importance of topographic details in numerical simulations. With a resolution of 1 arc-minute in the
ETOPO1 global dataset, the simulation of ISWs requires more details of topographic characteristics.
(3) Although tidal forcing cannot change the generation mechanism of ISWs, it has a modulating effect on the generation and
evolution of ISWs. During spring tide, ISWs are enhanced by a stronger tidal flow and the modulating effect of internal wave
beams and exhibit rank-ordered wave packets. However, during neap tide, ISWs become nearly west-east symmetric and
exhibit a single wave.
(3) The effect of seasonal stratification is negligible because of the small difference between summer and winter in tropical
regions. However, the deeper isopycline in summer makes ISWs propagate faster than in winter, and it also slightly enhances
the amplitude and baroclinic energy of eastward rank-ordered ISWs.
**Acknowledgements**
This work was supported by the National Natural Science Foundation of China through grant 41876012, the National Basic
Research Program of China (973 Program) through grant 2017YFC1405605 and the Fundamental Research Funds for the
Central Universities. The manuscript has undergone English language editing before submission through Springer Nature
Author Services (SNAS, verification code: 020C-84FE-7F7C-A390-D7FB).

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
