# Peer review of "On the Generation and Evolution of Internal Solitary Waves in the Andaman Sea"

_Nonlinear Processes in Geophysics, 2021_

## Referee Comment (RC1)

REVIEW

of the manuscript
"On the generation and evolution of internal solitary waves in the Andaman Sea" by
Yujun Yu, Jinhu Wang, Shuya Wang, Qun Li, Xu Chen, Jing Meng, Kexiao Lu and
Guixia Wang

The paper focuses on the problem of generation and evolution of internal waves in the Andaman Sea, specifically, their origin due to interaction of tidal flow with a ridge in the Nicobar Archipelago and successive evolution of generated waves in the surrounding areas. The problem is addressed by considering the model results obtained based on two-dimensional numerical modelling using the MITgcm. An interesting element of this paper is a contribution of two small ridges to the wave fields generated by the main ridge located between them. Some sensitivity runs on the temporal variability of the wave characteristics over the neap-spring cycle, as well as due to seasonal variations of the stratification were conducted. The influence of the variation of the bathymetry (different transects across the ridge) was also in the focus.

There are some interesting elements in the paper that deserve to be considered, although I personally have some doubts concerning the robustness of the applied method. It is quite difficult to judge the paper if you are not convinced in plausibility of the reported results. So, in my opinion the paper should not be accepted for the publication in the form it is submitted now. I would recommend the authors to conduct a much more detailed analysis taking some recommendations onboard which could be as follows.

A) I see three principal methodological flaws that concern the model setting. My first concern is about the two-dimensional replication of the internal tides generation in the Andaman Sea. In terms tidal energy conversion, the area of Andaman and Nicobar Islands is a cluster of point sources of generation in which waves radiate from every particular source radially. This is clearly seen in SAR images depicted in Fig.1 Even far away from the generation points the fronts of internal solitary waves are still curved presenting their three-dimensional origin. In terms of modelling, an attempt to replicate such a complex three-dimensional generation and evolution using a two-dimensional approach leads to a great overestimate of the amplitudes/energy of propagating waves.

B) Of course, the model setting can be tuned, and model predictions can be corrected based on the model validation which assumes comparison the model predicted wave characteristics with observational data. This is not the case of the present paper, and this is my second concern. In many cases the consistency of the observational data with the model output gives a confidence that the predicted wave characteristics reflect elements of the real

wave dynamics. However, I did not find in the text any attempt to justify the modelling results comparing them with observations. The paper is built on the "trust me" basis.

C) The lack of model validation put a great question mark on the robustness of the predicted wave characteristics. Specifically, with such a coarse grid taken in model setting, $\Delta z$=500m horizontal step, and $H_p$~90m thickness of the surface layer, Fig. 4c, the leptic ratio $\Delta z/ H_p$=~5.5 which is >> 1. In such conditions I expect predominance of the numerical dispersion over physical, and all results related to internal solitary waves shown in Figs. 8, 10, 13, 15 can be artifacts generated by the model with great numerical dispersion, but not actual physical phenomenon. Much finer model resolution is required for making numerical dispersion at least at the level or much below physical dispersion.

---

## Referee Comment (RC2)

*Review of*
**"On the generation and evolution of internal solitary waves in the Andaman Sea**
*by Y. Yu, J. Wang, S. Want, Q. Li, X. Chen, J. Meng, K. Lu and G. Wang*

This paper reports on two-dimensional numerical simulations of the generation of internal solitary waves by tidal currents across a ridge in the Nicobar Archipelago. The focus is on the impact of two small ridges superimposed on either side of the main ridge. Differences in the generation of ISWs during spring and neap tides and between winter and summer are also discussed. The experiments using different bathymetries are the potentially useful contribution. Comparing spring and neap tides or winter and summer stratifications not so much.

Unfortunately a horizontal resolution of only 500 m was used which seems too low to resolve ISWs. So I am unconvinced by these simulations. Are the ISWs due to numerical dispersion or physical dispersion? That numerical dispersion can result in the formation of ISWs was discussed in Vitousek and Fringer (Ocean Modelling, 2011). Evidence that the simulated ISWs are physical is required. I suggest the authors do a simulation with a resolution of, say, 50 m to check. There should be no reason that this can't be done as the MITgcm runs very efficiently on large numbers of cores. For this reason I recommend that the paper be rejected but I encourage the authors to resubmit after more careful simulations have been done as there are potentially some useful results.

It is difficult to comment on the results (e.g., the comparisons in Figure 10) without first being convinced that they are real so I will leave that to a future version of the paper. So I only have a few minor comments.

Comments:

1. Line 51. The $10°$ Channel is the name of a channel in this region but this is not made clear. I think the proper name may be 'Ten Degree Channel'. Also 'secondary ISWs' should be explained.

2. Line 58. 'modal' should be 'model'.

3. Line 87. Why say that S1 and S2 'may' have an affect on the generation of ISWs?

4. Line 118. 'tidal current'. When $\sigma \ll 1$ you generate higher harmonics as well.

5. Line 119. 'Lee' should not be capitalized here or elsewhere.

6. Lines 127–128. The topography always affects the flow regardless of the value of $Fr_t$! How can you possibly say the the flow is not affected by the topography when $Fr_t > 1$?

7. Table 3. Explain what $L$ and $H_{R0}$ are. What is $c_1$? On line 131 you have $c$ and $c_i$ in the formula.

8. Line 157. Do the generation points coincide with critical slopes of regions of large IBF?

9. Line 184. 'rank'

10. Equation (6). This is a linear approximation for the available potential energy. Why not use a correct formula, particularly as you are using this to compute the energy of solitary waves.

11. Line 192. 'Bengal'

12. Line 200. Why 'may be related'? Is is related or not?

13. Line 218. Which direction is ebb? It would be better to use leftward and rightward as simpler to connect to the figures.

14. Line 230. Is Exp4 or Exp3 similar to the realistic topography (the way this is written is not clear although of course it is from the figures).

15. Line 264. Exp8 is not in table 2. What is Exp8?

16. Line 295. 'pycnocline'